# Formation and Characterization of Chitosan-Based Polyelectrolyte Complex Containing Antifungal Phenylpropanoids

**DOI:** 10.3390/polym16233348

**Published:** 2024-11-29

**Authors:** Andrés F. Olea, Héctor Carrasco, Franco Santana, Laura Navarro, Raúl Guajardo-Maturana, Cristian Linares-Flores, Nancy Alvarado

**Affiliations:** 1Grupo QBAB, Instituto de Ciencias Aplicadas, Facultad de Ingeniería, Universidad Autónoma de Chile, El Llano Subercaseaux 2801, San Miguel, Santiago 8910060, Chile; andres.olea@uautonoma.cl (A.F.O.); hector.carrasco@uautonoma.cl (H.C.); 2Carrera de Ingeniería Civil Química, Facultad de Ingeniería, Universidad Autónoma de Chile, Avenida Pedro de Valdivia 425, Providencia, Santiago 7500912, Chile; franco.santana1@cloud.uautonoma.cl; 3Instituto de Ciencias Biomédicas, Facultad de Salud, Universidad Autónoma de Chile, El Llano Subercaseaux 2801, San Miguel, Santiago 8910060, Chile; laura.navarro@uautonoma.cl; 4Instituto de Investigación Interdisciplinar en Ciencias Biomédicas SEK (I3CBSEK) Chile, Facultad de Ciencias de la Salud, Universidad SEK, Fernando Manterola 0789, Providencia, Santiago 7520317, Chile; raul.guajardo@usek.cl; 5Instituto de Ciencias Naturales, Universidad de Las Américas, Manuel Montt 948, Santiago 7500975, Chile; cris.linares.flores@gmail.com; 6Facultad de Ingeniería, Arquitectura y Diseño, Universidad San Sebastián, Bellavista 7, Santiago 8420524, Chile

**Keywords:** chitosan, anhydride maleic derivative, encapsulation, active compounds

## Abstract

In this work, a novel chitosan-based polyelectrolyte complex (PEC) was prepared using chitosan as the cationic polyelectrolyte, while a potassium salt of poly(maleic anhydride-alt-tetradecene) (PMA-14) served as the anionic counterpart. These PECs were used for the encapsulation of two nitroeugenol derivatives: 4-allyl-2-methoxy-6-nitrophenol (**3**) and 2-allyl-6-nitrophenol (**4**). The results confirm complex formation and efficient encapsulation of active compounds. Encapsulation efficiency (EE) was influenced by the chemical structure of the compounds, with 32.18% EE for **3** and 20.36% EE for **4**. The resulting systems were characterized by fluorescence probing techniques, dynamic light scattering (DLS), and zeta potential. On the other hand, antifungal assays revealed that, in free form, **3** exhibits a much higher activity against *Botritys cinerea* than **4**. However, no effect of encapsulation of both compounds on antifungal performance was observed. Results from molecular dynamic studies indicate that a stabilization effect is induced by compounds **3** and **4** during PEC formation, which is attributed to specific interactions between polyelectrolytes and guest molecules. These results are in line with the EE values measured for **3** and **4** and explain the low release from PECs of these molecules. Thus, the potential development of PEC-based systems for the delivery of bioactive compounds requires a deeper comprehension of parameters determining the relationship between encapsulation efficiency and delivery kinetics.

## 1. Introduction

Chitosan is a versatile biopolymer known for its cationic nature, biocompatibility, biodegradability, and structural flexibility [1,2,3,4]. Its amino and hydroxyl groups enable strong hydrogen bonding and electrostatic interactions, allowing chitosan to form nanoaggregates resulting in stable and functional materials suitable for a wide range of applications [5,6,7].

Polymeric nanoaggregates have emerged as key components in the field of encapsulation and transportation, owing to their unique properties and characteristics. These nanostructures, typically ranging from 1 to 100 nanometers, exhibit a high surface area-to-volume ratio, a feature that proves essential in encapsulation applications [8,9,10]. One notable example is the utilization of amphiphilic block copolymers as nanoaggregates, which can form self-assembled structures capable of encapsulating hydrophobic drug molecules within their core. The amphiphilic nature of these polymers allows them to adapt to complex biological environments, making them ideal candidates for drug delivery systems [11]. Moreover, the adjustable nature of polymeric nanoaggregates enables the precise tailoring of their properties for specific applications. Among these nanostructures, polyelectrolyte complexes (PECs) have emerged as a remarkable class of materials, gaining considerable attention in the scientific community for their adaptable characteristics and numerous applications [12,13]. These complexes are formed through the intricate interplay of charged polymers, with oppositely charged components coming together via electrostatic interactions. Such assemblies exhibit a notable array of structural morphologies, ranging from simple nanoparticles to more complex multilayered films, providing researchers with a rich array for tailoring material characteristics.

The choice of polyelectrolyte pairs is a critical factor that influences the stability, structure, and properties of the resulting complexes. Thus, various combinations of positively charged (cationic) and negatively charged (anionic) polyelectrolytes are possible, such as poly(allylamine hydrochloride) (PAH) and sodium poly(styrene sulfonate) (PSS) [14,15]; poly(diallyldimethylammonium chloride) (PDADMAC) and poly(sodium 4-styrenesulfonate) (PSS) [16,17,18]; chitosan and sodium alginate [19,20,21]; poly(ethyleneimine) (PEI) and poly(acrylic acid) (PAA) [22,23,24]; and poly(L-lysine) (PLL) and hyaluronic acid (HA) [25,26]. These examples demonstrate the range of polyelectrolyte combinations used in the preparation of PECs, which requires the careful selection of polymer pairs. The formation of PECs involves a series of factors that control the assembly process, including polymer concentration, molecular weight, and ionic strength of the surrounding medium [12]. The resulting structures exhibit a remarkable diversity that ranges from nanoscale particles [27,28,29] and microcapsules [30,31] to thin films [32,33], each holding distinct potential for practical applications. The ability to control size, composition, and surface characteristics is key to enhancing their biocompatibility and stability, crucial factors for effective encapsulation and transportation.

Phenylpropanoids, such as eugenol (4-allyl-2-methoxyphenol) (**1**), a natural phenolic compound primarily derived from *Syzygium aromaticum*, and 2-allylphenol (**2**), a biomimetic compound that emulates ginkol, isolated from *ginko biloba*, have garnered recognition across various disciplines due to their exceptional qualities (Figure 1). These compounds exhibit several biological activities, highlighting among them antimicrobial, anti-inflammatory, antioxidant, and antifungal properties [34,35,36,37,38]. Chemical modifications of phenylpropanoids have been carried out to enhance these activities and, at the same time, have allowed the proposal of some structure–activity relationships that could be useful for potential applications of these compounds [39,40,41]. For example, it has been shown that attachment of nitro groups to the aromatic ring of **1** increases the mycelial growth inhibition of *Botrytis cinerea*, whereas the same kind of substitution on **2** decreases the growth inhibition of *Phytophthora cinnamomic*. However, despite their remarkable properties, in general, the active compounds often face challenges such as poor stability, low bioavailability, and degradation due to environmental or medium factors like light, oxygen, moisture, pH, or temperature [42,43,44]. In this way, encapsulation offers a potential solution by protecting these compounds for improving these barriers [45,46,47,48]. In this context, PECs appear as an attractive alternative for the encapsulation of these active, mostly hydrophobic compounds; thus, the solubility of active compounds can be enhanced, facilitating their administration or application in different media [8,49,50].

In this study, PECs were formed using chitosan as the positively charged polyelectrolyte (Figure 2A) because this biopolymer offers key advantages to form polyelectrolyte complexes, such as biocompatibility, biodegradability, and minimum environmental impact. As a counterpart, the potassium salt of poly(maleic anhydride-alt-tetradecene) (PMA-14) is used as an anionic polyelectrolyte because their behavior in aqueous solution is well established [51] (Figure 2B). These PECs were developed to encapsulate 4-allyl-2-methoxy-6-nitrophenol (**3**) and 2-allyl-6-nitrophenol (**4**) (Figure 1), two nitro derivatives obtained from eugenol and 2-allylphenol, respectively. These encapsulated compounds were used to assess the influence of PECs on their antifungal activities. The mycelial growth inhibition of *B. cinerea* was evaluated in the presence of both free and encapsulated forms of compounds **3** and **4**. Results are discussed in terms of PEC physicochemical properties, with molecular dynamics simulations providing further insights into the discussion.

This study addresses the critical need to understand the factors influencing the formation and morphology of chitosan-based PECs. Parameters such as particle size, surface charge, polymer configuration, alkyl chain length, and the presence of non-polar guest molecules were studied, offering valuable insights for tailoring these complexes to enhance their functionality and optimize their application as drug carriers.

## 2. Materials and Methods

Chitosan (MW: 310,000–375,000 Da; deacetylation degree > 75%), poly(maleic anhydride-alt-tetradecene) (Mn: 7300 Da; Mw: 9000 Da), eugenol, and 2-allylphenol were purchased from Aldrich (St. Louis, MO, USA) and used without purification; potassium hydrogen sulfate, sodium nitrate, sodium hydroxide, Na_2_SO_4_, K_2_CO_3_, silica gel, hydrochloric acid 37%, glacial acetic acid 100%, dichloromethane, and ethyl acetate were obtained from Merck (Darmstadt, Germany); and potato dextrose agar, hexane, and methanol were purchased from JT Baker (Alcobendas, Madrid, Spain).

### 2.1. Phenylpropanoids Derivatives

Phenylpropanoid derivatives **3** and **4** were obtained from eugenol (**1**) and 2-allylphenol (**2**), respectively. Briefly, compounds **1** or **2** are treated with sodium nitrate and potassium hydrogen sulfate in dichloromethane at room temperature following a previously reported procedure [37]. Reaction products were purified by column chromatography and characterized using ^1^H and ^13^C RMN mono and bi-dimensional spectroscopy and compared with standard samples previously prepared [37].

### 2.2. Salt Formation of Poly(Maleic Anhydride-Alt-Tetradecene)

Poly(maleic anhydride-alt-tetradecene) (1 g) was dissolved in hot KOH aqueous solution (30% *w*/*w*) and stirred for 24 h. Subsequently, the solution was precipitated by pouring on cold methanol. The resulting product was filtered and dried at 35 °C. Following this, the polymeric salt (PMA-14) underwent dialysis in a 3.5 kDa membrane bag until its conductivity matched that of the milli-Q water used. Finally, PMA-14 solution was freeze-dried.

### 2.3. PEC and PEC-Loaded Preparation

Chitosan was dissolved in 2% acetic acid (*w*/*w*) to achieve a 1.55 × 10^−2^ M concentration (repetitive unit basis), while PMA-14 was dissolved in milli-Q water to obtain a concentration of 5.6 × 10^−3^ M. This chitosan solution was loaded into a syringe pump and dispensed onto PMA-14 solution at a rate of 1 mL/h under constant stirring. The mixture was stirred for 24 h at room temperature. The volumes of chitosan and polymeric salt used were adjusted in such a way that 50% of chitosan charges became neutralized, and therefore, a PEC with a resulting positive charge was obtained.

Incorporation of phenylpropanoid derivatives (**3** and **4**) into PEC was achieved as follows. Briefly, predetermined amounts of cited compounds (to achieve a final concentration of 1 mg/mL) were incorporated during the PEC formation process. The mixture was continuously stirred at room temperature for 24 h. Following centrifugation, the supernatant containing PEC with encapsulated **3** and **4** derivatives was collected for further analysis.

### 2.4. Dynamic Light Scattering (DLS) and Zeta Potential

Size distribution and zeta-potential (ζ potential) of PECs and their polyelectrolyte components were determined by dynamic light scattering (DLS) or Electrophoretic Light Scattering (ELS), respectively, using a Zetasizer Lab Blue (Malvern Instruments, Malvern, UK). The samples were placed into polystyrene or DTS1070 cuvettes for DLS and ELS, respectively.

### 2.5. Fluorescence Analysis

Steady-state fluorescence emission spectra of pyrene were recorded at room temperature by exciting at 335 nm using a Fluoromax-4 spectrofluorometer (Horiba Jobin Yvon, Piscataway, NY, USA). The emission range scanned was from 350 to 450 nm with an excitation wavelength of λex = 335 nm and an excitation slit width of 2.5 nm. The ratio of intensities of the first and third vibronic bands, I_1_/I_3_, serves as micropolarity indicator of the media sensed by the pyrene molecule [52].

### 2.6. Theoretical Characterization

Interactive structures that arise from encapsulation of active compounds in chitosan–polymer system were modeled using the ReaxFF code incorporated in the SCM package [53]. A geometry equilibration was reached by using a step-by-step simulation process. Molecular dynamics (MDs) allow for obtaining the most stable conformational rearrangements according to the involved interactions in the system. The reactive force field (ReaxFF) was selected, giving special relevance to the substrate chitosan–polymer interactions [54]. The lattices of the modeling box were fixed at 30 Å, which permits free movement, where the starting configuration of higher energy consists of compounds placed at enough distance among them, which quickly evolved to more energy-favorable positions due to attracting interactions. The simulation was performed in an NTV ensemble without any constraints, and the data collection was saved every 12.5 fs; the step value was 0.25 fs, with a total simulation time of 100 ps, running 4 × 10^5^ computational steps. The temperature was equilibrated within Berendsen’s thermostat, and the mean Verlet algorithm solved the motion equations (Figure 3).

### 2.7. Encapsulation Efficiency (EE) and Loading Capacity (LC)

A specific volume of PEC loaded with compounds 3 and 4, as described in Section 2.3, was used to measure absorbance. Along with the above, a calibration curve for **3** and **4** was first prepared using dichloromethane solutions from 0.05 to 0.28 mg/L. Absorbance was measured at 302 nm with dichloromethane as the blank. The obtained absorbance values were then converted to concentrations using the equation of the curve calibration. Subsequently, encapsulation efficiency and loading capacity were calculated using Equations 1 and 2, respectively.
(1)$$EE\%=\frac{m_{Encap}}{m_{Total}}\times 100$$
(2)$$LC\%=\frac{m_{Encap}}{m_{PEC}}\times 100$$
where m_Encap_ and m_Total_ denote the encapsulated mass and total mass of **3** or **4**, respectively; and m_PEC_ represents the total PEC mass used.

### 2.8. Antifungal Activity Against Botrytis cinerea

To determine the inhibitory activity of PEC components and compounds **3** and **4** on *B. cinerea*, the phytopathogen was cultured on potato dextrose agar (PDA) in 90 mm Petri dishes for 10 days. Subsequently, PDA plates were prepared with 100 mg/L of each polyelectrolyte or active compound, and 6 mm diameter mycelial discs of *B. cinerea* from the pregrown culture were placed at the plate center. These plates were incubated for ten days at 21 °C, and the diameter of the fungal growth was measured every 48 h. All assays were performed in triplicate.

## 3. Results and Discussion

### 3.1. Polyelectrolyte Complex Formation

Polyelectrolyte complexes were formed by mixing chitosan, as cationic polyelectrolyte, and potassium salt of poly(maleic anhydride-alt-tetradecene) PMA-14, as anionic polyelectrolyte. The formation of the chitosan/PMA-14 complex was monitored by fluorescence probing technique and dynamic light scattering (DLS) measurements.

### 3.2. Fluorescence Probing

Fluorescence techniques have been used to monitor self-aggregation, conformational changes, and binding dynamics that involve polyelectrolytes, amphiphilic block copolymers, and PEC [55,56,57,58]. Additionally, pyrene has been used as a fluorescent probe to sense hydrophobic environments because its fluorescence spectrum exhibits characteristic shifts and intensity changes based on the polarity of its surroundings [52,59]. Therefore, pyrene fluorescence spectra in aqueous solution and the presence of PMA-14 or chitosan/PMA-14 complexes were obtained. The spectra and a schematic representation are shown in Figure 4 and Figure 5.

Interestingly, pyrene fluorescence spectra (B and C) obtained in the presence of PMA-14 and chitosan/PMA-14 complexes show two main features compared with pyrene spectrum registered in aqueous solution (A), i.e., fluorescence vibronic bands shift to higher wavelengths (redshift), and the ratio of intensities of the first (I_1_) and third (I_3_) vibronic bands (I_1_/I_3_) decreases. More specifically, in an aqueous solution, the ratio of I_1_/I_3_ is 1.7, whereas in the presence of PMA-14 and chitosan/PMA-14 complexes, this value decreases to 1.18 and 1.09, respectively. It is well established that the ratio I_1_/I_3_ is strongly dependent on the environmental polarity sensed by pyrene [52]. Based on this fact, a solvent polarity scale has been proposed [59], i.e., a high value of the ratio I_1_/I_3_ indicates a polar environment, whereas the lowest values of the I_1_/I_3_ ratio indicate less polar media (more hydrophobic). Therefore, these results indicate that the formation of PEC generates a less polar microdomain than that provided by the coiling of PMA-14. It has been previously reported that hydrophobic interaction between long hydrocarbon side chains of PMA-14 induces the formation of compact coils, in which pyrene exhibits I_1_/I_3_ equal to 1.17 [60]. This hydrophobic effect should also be present after the formation of a polymer complex between oppositely charged chains of chitosan and PMA-14. However, the lowest ratio of I_1_/I_3_ suggests that interaction between alkyl side chains of PMA-14 is stronger in the PEC, leading to tightly packed hydrophobic domains in which pyrene is shielded from the surrounding aqueous environment. The PEC surface has 50% of the charged groups remaining, and these are responsible for PEC water solubility. On the other hand, the neutralized chitosan groups are probably pushed to the inner non-polar microdomain formed by alkyl side chains of PMA-14. This effect could play a role in the formation of a tighter environment inside the PEC. Consequently, it is expected that the chitosan/PMA-14 complex possesses large and tight hydrophobic domains, in which pyrene senses a less polar environment than in PMA-14 coils.

### 3.3. Size Distribution and Zeta Potential (ζ)

Dynamic light scattering (DLS) is a versatile technique that is widely used to determine the size distribution of particles dispersed in different media. Thus, DLS measurements in aqueous solution were carried out using a Zetasizer Lab Blue, which determines fluctuations in scattered light intensity caused by the Brownian motion of particles in solution, providing directly the hydrodynamic diameter in aqueous solution. The plots of intensity against diameter obtained for the chitosan/PMA-14 complex and both polyelectrolytes are shown in Figure 6.

DLS measurements for chitosan and copolymer PMA-14 show one size distribution for each macromolecule, with average hydrodynamic sizes of 6 nm and 120 nm, respectively. Interestingly, the hydrodynamic diameter of PMA-14 is much larger than that found for chitosan. This size difference can be attributed to the conformation adopted by these polyelectrolytes in aqueous solution. Chitosan is a positively charged linear polymer, whereas the copolymer PMA-14 forms intramicellar aggregates in the whole range of pH [51,60]. On the other hand, the DLS of the chitosan/PMA-14 complex gives a bimodal size distribution with average hydrodynamic sizes of 660 nm and 4700 nm. This result indicates the existence of a heterogeneous system in which two particle populations of different sizes coexist. Probably, the highest-size population is formed by higher-order structures resulting from the aggregation of small and spherical PEC.

Additionally, zeta potential (ζ) values for PEC and its polymer components were obtained by ELS measurements carried out in the same Zetasizer instrument. Zeta potential is a measure of repulsion between particles with the same charge. These repulsive forces contribute to avoiding particle aggregation, and therefore, ζ is commonly associated with the physical stability of aqueous suspension of charged particles [61]. Namely, ζ ≥ ±30 mV is indicative of stable suspensions. The obtained values for chitosan, PMA-14, and PEC for ζ are +88, −40 mV, and +75 mV, respectively. The resulting positive charge in the chitosan/PMA-14 complex is in line with the non-stoichiometric quantities of negative and positive charges used in PEC preparation (n_−_/n_+_ = 0.36), i.e., excess of positive charges on chitosan chains is present in the final mixture. It is important to emphasize that in the PEC preparation, the initial solution contains the anionic polyelectrolyte PMA-14, which is deficient in the mixture, and chitosan is slowly added drop by drop. In this process, it is expected that primary complexes are formed until PMA-14 chains are completely bound to chitosan chains, and then the excess of chitosan interacts with the initially formed PEC, forming a positively charged surface [62].

These results indicate that the chitosan/PMA-14 complex is formed by a hydrophobic inner core surrounded by cationic chitosan chains. This point has been pointed out in a study of PEC formation between poly(diallyldimethylammonium chloride) (PDADMAC) and sodium salt of poly(maleic anhydride-alt-propene), PMA-3 [62]. However, the spatial arrangement of cationic and anionic polyelectrolytes should be completely different. In the PDMADMAC/PMA-3 complex, the interaction between the positive and negative polymers leads to almost stoichiometric neutralization because the distance between charges on both monomer units is quite similar. On the other hand, positive groups on chitosan have the largest charge distances and different spatial orientations (equatorial upside or downside). Consequently, the ionic interaction between monomer units of chitosan and PMA-14 cannot be 1:1 because it occurs on only one of these two orientations. In other words, the interaction between polymer chains is not strictly linear, and a degree of entanglement must exist. In this configuration, alkyl chains of PMA-14 must play an important role in the final PEC configuration and size of the PEC. For example, the reported size for PDMADMAC/PMA-3 formed under a similar ratio of negative and positive charges is 250 nm, whereas for chitosan/PMA-14, the size increases to 660 nm.

Interestingly, the presence of loading molecules in the PEC formation process induces changes in both parameters. On one side, the size distribution obtained in the presence of compound **3** gives rise to just one size distribution centered around 4040 nm (Figure 7), and a similar pattern occurs with compound **4**, showing a peak around 4700 nm. Meanwhile, ζ measurements indicate that the zeta potential decreases when **3** and **4** are loaded in PEC (Figure 8).

Results in Figure 7 suggest that the presence of hydrophobic molecules somehow affects the mechanism of complexation of these opposite-charged polyelectrolytes. A possibility is that the formation of the chitosan/PMA-14 complex and incorporation of these molecules to the intramolecular micellar media provided by PMA-14 copolymer occurs simultaneously, and then the internal microdomain is adapted to accommodate guest molecules in the coiled PMA-14 chains. In this sense, the larger size of PEC/4 compared with PEC/3 would be due to the tendency of compound **4** to position itself at the edges of the PEC structure, resulting in an overall larger PEC size than with compound **3**. This suggests that compound **3** may interact more effectively with the PEC.

On the other hand, the decrease in ζ in the presence of **3** must be related to the disappearance of PEC with the lowest size distribution. In other words, as the PEC size increases and the positive charge number is the same, the charge surface density decreases. It is worth mentioning that zeta potential is obtained by measuring the electrophoretic mobility of charged particles in an applied electric field. Since mobility is influenced by size and charge, zeta potential values may reveal changes in both parameters due to changes in the PEC formation mechanism and by the adsorption of charged species onto the complex surface.

In summary, for PEC formed by chitosan and PMA-14, both particle size and surface charge are crucial factors for drug carrier applications, depending on the polymer configuration resulting from the ionic interaction and probably on the alkyl chain length as well. Additionally, the presence of non-polar guest molecules also has an important effect on these parameters.

Further investigation into the factors governing complex formation and morphology could provide valuable insights into this mechanism for tailoring physicochemical properties of chitosan-based PECs for targeted applications.

### 3.4. Encapsulation Efficiency (EE)

Encapsulation efficiency (EE) is a parameter of paramount importance for the development of PEC applications, such as drug delivery, tissue engineering, and biosensing. EE dictates dosage accuracy, minimizes waste, protects the encapsulated cargo, influences its release profile, and allows for optimization of PEC formulations, ensuring controlled and efficient delivery. Herein, EEs were calculated according to Eqn. 1, and values of 32.18% and 20.36% were obtained for compounds **3** and **4**, respectively. This variation in EE may be attributed to different interactions of guest molecules with the PEC matrix, which, in its term, is determined by their chemical structure. As seen in Figure 1, the main difference between these compounds is that compound **3** possesses an extra methoxy group attached to the aromatic ring. This methoxy group in **3**, being an electron-donating group, likely increases the electron density of the nitro group through resonance. This enhanced electron density could strengthen the electrostatic interactions between the nitro group and the positively charged chitosan chains, leading to more efficient encapsulation within the complex. Additionally, the methoxy group may participate in hydrogen bonding with the amine and hydroxyl groups of chitosan, further stabilizing the encapsulation process.

In contrast, compound **4**, which lacks the methoxy group, may experience weaker electrostatic interactions with chitosan due to the reduced electron density of its nitro group. Additionally, the absence of potential hydrogen bonding interactions could result in less efficient encapsulation compared with compound **3**. Furthermore, the methoxy group in compound **3** might introduce steric hindrance that alters the molecule’s conformation, potentially facilitating a more favorable orientation for interaction with the polyelectrolyte matrix. This conformational effect could also contribute to the observed higher encapsulation efficiency for compound **3**. The measured loading capacity values were 8.74% for PEC/3 and 5.53% for PEC/4.

These findings emphasize the significance of subtle structural modifications in active compounds for modulating their encapsulation within polyelectrolyte complexes. By understanding the interplay between functional groups, electronic effects, and steric factors, it becomes possible to rationally design encapsulation systems with enhanced efficiency and tailored release profiles, opening new avenues for drug delivery, food preservation, and other applications.

### 3.5. Antifungal Activity

The antifungal activity of chitosan has been widely reported on human and plant pathogens [63], and considering the biodegradability and safety of the compound, many derivatives have been synthesized to improve its natural antimicrobial properties [64]. On the other hand, natural phenylpropanoids and derivatives have also shown antifungal activity against several phytopathogenic fungi [39,40,41,65]. Thus, to assess the effect of encapsulation of compounds **3** and **4** in PEC formed by chitosan and PMA-14, the mycelial growth inhibition against *Botrytis cinerea* of each component of this mixture was evaluated (Figure 9).

The data indicate that from both PEC components, only chitosan exhibits antifungal activity, whereas derivative **3** is much more active than **4** and shows the most potent inhibitory effect. Interestingly, the inhibition effect of derivative **3** encapsulated in the PEC (PEC/3) is like that shown by chitosan, suggesting that *B. cinerea* is only affected by the polymer complex, and **3** is unable to reach the fungus. In other words, the encapsulated compounds remain trapped by the PEC and therefore have no effect on mycelial growth. This effect could be attributed to the strong interaction between derivatives and forming complex polyelectrolytes, which leads to high encapsulation efficiency but extremely low bioavailability. Thus, even though **3** and **4** are present in the growing fungus media, these molecules cannot interact with *B. cinerea* because they are buried in the PEC. Consequently, the observed growing inhibition effect is exclusively due to chitosan, which forms the PEC external surface.

These findings suggest that it is mandatory to understand the interplay between the chemical structure of guest molecules and their interactions with PEC to design encapsulation systems with good loading capacity and tailored release profiles. With this aim, a molecular dynamic study was performed.

### 3.6. Theoretical Analysis

The incorporation of derivatives **3** and **4** into the microdomains provided by the chitosan-PMA-14 complex is mainly determined by the interactions between the constituent polyelectrolytes and the active compounds. To gain deeper insights into these interactions, molecular dynamic (MD) simulations, replicating real-world conditions, were performed. Thus, a 30 Å box was constructed, incorporating the polyelectrolytes alongside their respective counterions to maintain charge neutrality and the active compounds. Also, to be aligned with experimental conditions, an explicit aqueous environment was added. During the MD simulation of loaded PEC, notable rearrangements of chitosan and PMA-14 were observed (Figure 10).

These rearrangements are due mainly to electrostatic forces driving polyelectrolytes together, minimizing spatial volume and leading to the lowest energy configuration. In this process, at the lowest extent, the molecules that are incorporated into the complex structure participate as well. These changes and interactions between derivatives **3** and **4** with both chitosan and PMA-14 chains are illustrated in Figure 10.

The calculated energy changes occurring during the transformation, which takes the system from the initial to the final configuration, are −234 kcal·mol^−1^ and −131 kcal·mol^−1^ in the presence of compounds **3** and **4**, respectively (see Table 1).

These results indicate that derivatives induce a stabilizing effect on PEC structure during the encapsulation process, i.e., a more negative energy difference is obtained, and the magnitude of this change is higher for **3** compared with that obtained with **4**. This means that the interaction of these molecules within the PEC structure is different. Figure 10 shows that in the final state, compound **3** interacts mainly with chitosan chains, whereas **4** is preferentially in contact with alkyl polymer chains of PMA-14. These different interactions and locations are due to chemical differences between these two derivatives. Even though both molecules have similar polarizabilities, the ability to form hydrogen bonds is completely different. In the case of compound **4,** the hydroxyl group and the alkyl side chain are in relative *ortho* positions, and therefore, the formation of H bonds could be prevented by the steric effect. On the other hand, in compound **3**, the hydroxyl group is free to form hydrogen bonds with the oxygen atoms of the chitosan. It is worth mentioning that these results are in line with the experimental determination of encapsulation efficiency, i.e., the EE value obtained for **3** is higher than that measured for **4**. However, the interaction of both molecules with the PEC is strong enough to avoid their release and, consequently, minimize their antifungal activity.

These results suggest that a deeper comprehension of parameters determining the relationship between encapsulation efficiency and delivery kinetics is needed.

## 4. Conclusions

The preparation and characterization of a chitosan-based polyelectrolyte complex (PEC) and the encapsulation of phenylpropanoid derivatives were performed. Fluorescence probing indicates the existence of distinct environments with different polarities, driven by hydrophobic and electrostatic interactions that shape PEC structural properties. Dynamic light scattering and zeta potential measurements revealed a bimodal particle distribution and positive surface charge. Experimental values of encapsulation efficiency indicate that **3** is more efficiently incorporated than **4**, and at the same time, no improvement in the antifungal activity of both encapsulated derivatives was observed. Molecular dynamic simulations suggest that these results are due to differences in the molecular structure of guest molecules, especially the distribution of substituent groups in the aromatic ring.

In summary, interactions between both derivatives and the PEC enhance the encapsulation process and likely cause loaded compounds to remain within the PEC environment, limiting their release and subsequent antifungal activity. These findings highlight the importance of a deeper understanding of the relationship between the chemical structure of guest compounds with encapsulation efficiency and effective delivery.

Results from this research pave the way for the development of PEC-based systems for the delivery of bioactive compounds, offering strategic opportunities for enhanced antifungal treatments and broader agricultural applications.

## Figures and Tables

**Figure 1 polymers-16-03348-f001:**
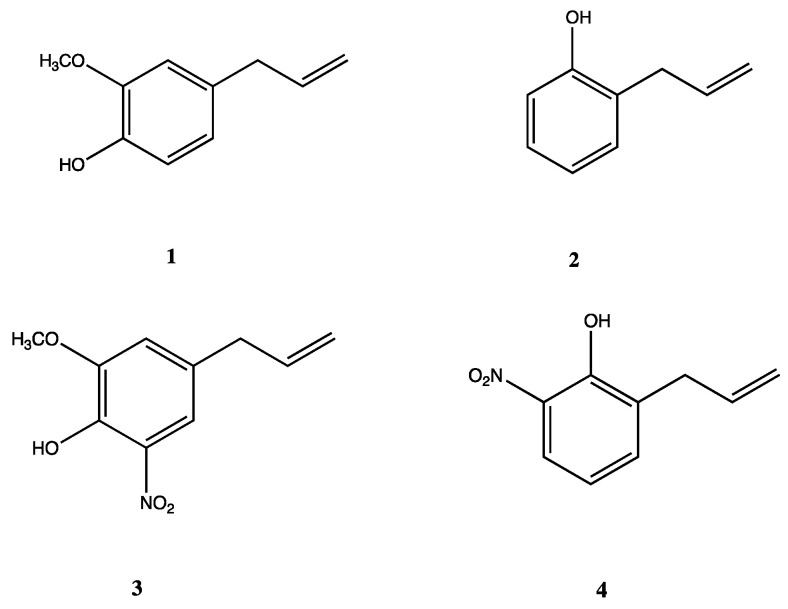
Eugenol (**1**) and synthetic phenylpropanoids: 2-allylphenol (**2**), 4-allyl-2-methoxy-6-nitrophenol (**3**), and 2-allyl-6-nitrophenol (**4**).

**Figure 2 polymers-16-03348-f002:**
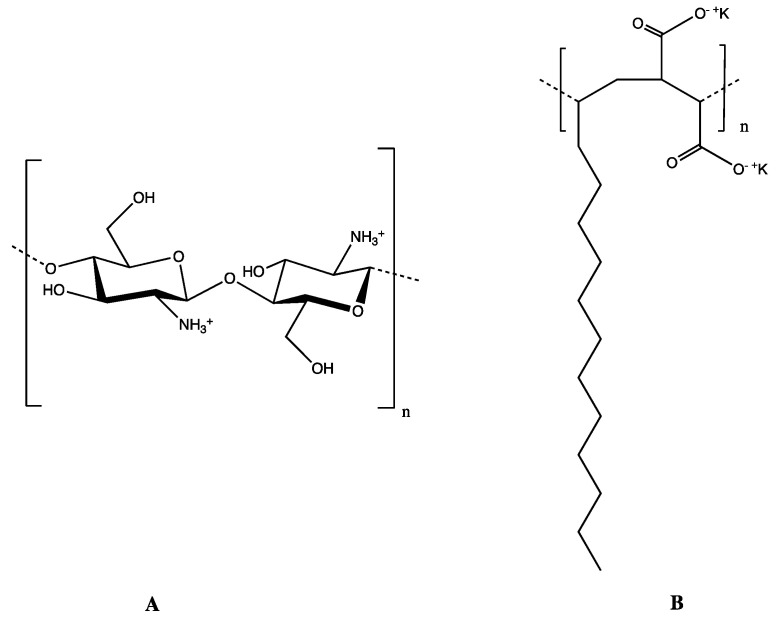
Polyelectrolytes used to form PEC: (**A**) chitosan and (**B**) potassium salt of poly(maleic anhydride-alt-tetradecene), PMA-14.

**Figure 3 polymers-16-03348-f003:**
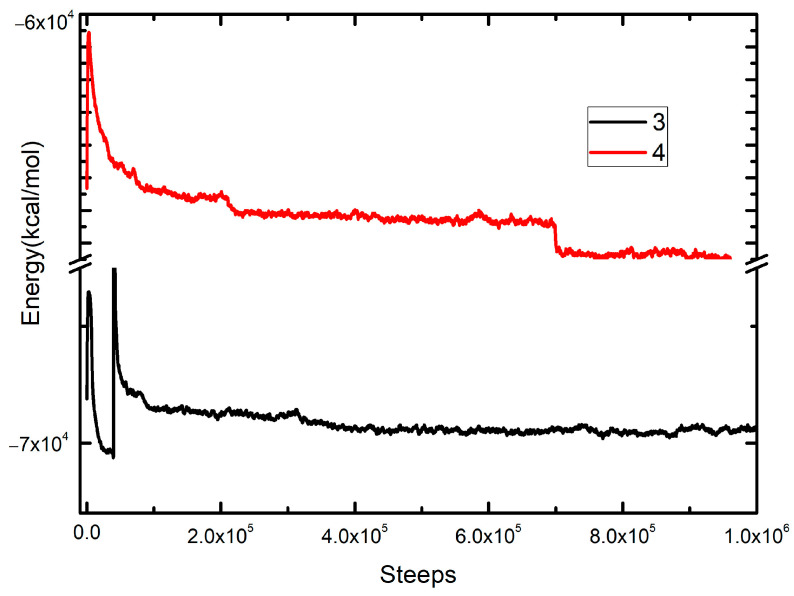
Molecular dynamic steps for chitosan/PMA-14 complex, including active compounds **3** and **4**.

**Figure 4 polymers-16-03348-f004:**
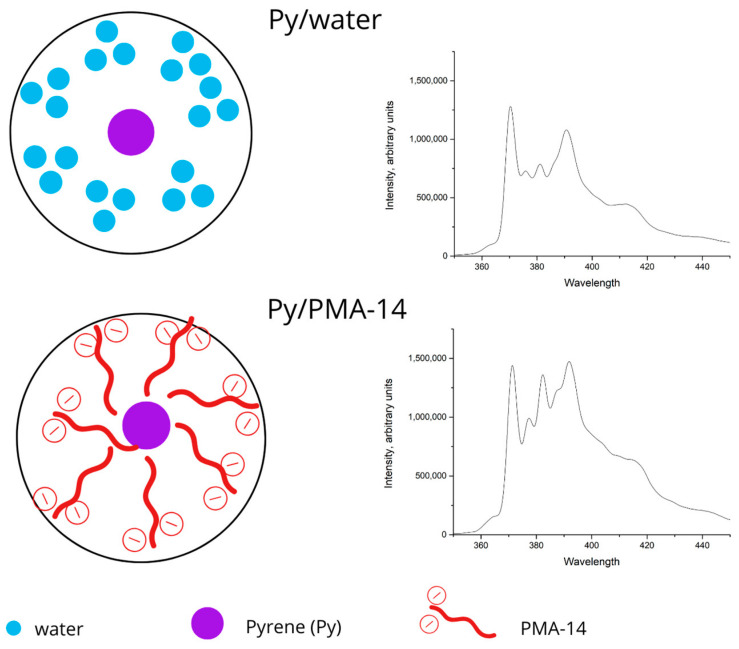
Schematic representation of pyrene in aqueous and PMA-14 environments.

**Figure 5 polymers-16-03348-f005:**
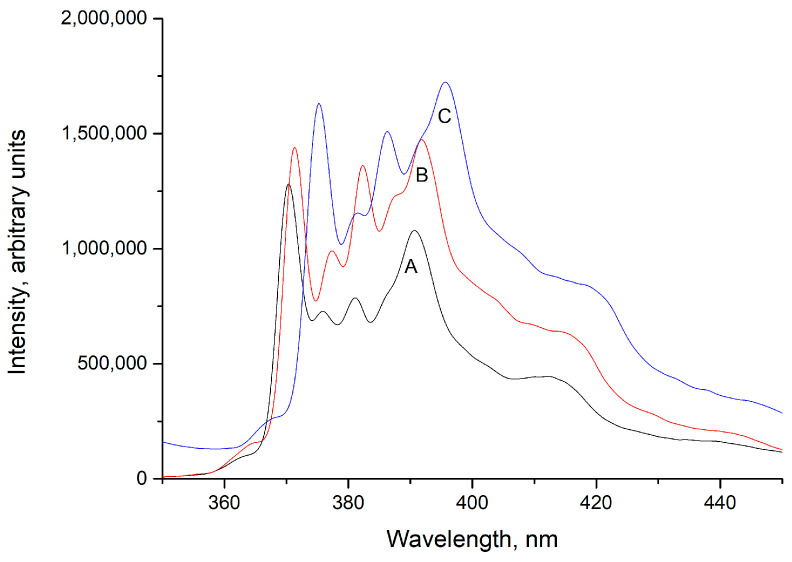
Fluorescence spectra of pyrene (A) in aqueous solution, (B) in presence of PMA-14 1mM, (C) in presence of chitosan/PMA-14 complex.

**Figure 6 polymers-16-03348-f006:**
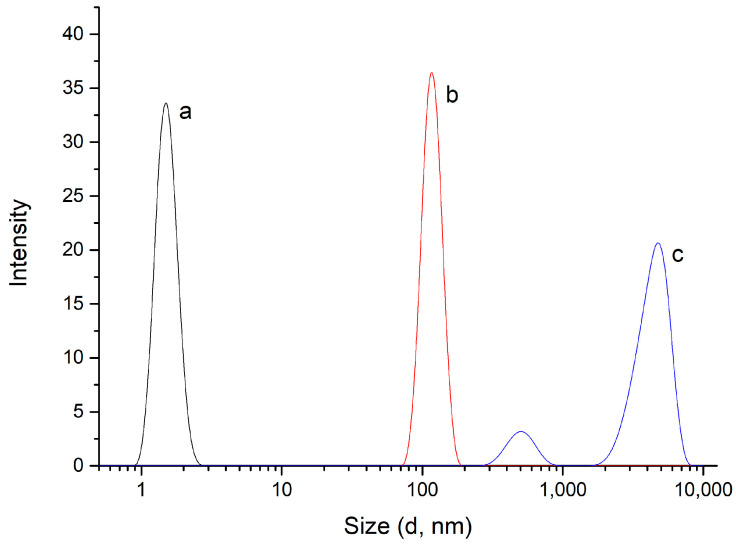
DLS measurements for (a) chitosan, (b) PMA-14, (c) chitosan/PMA-14 complex.

**Figure 7 polymers-16-03348-f007:**
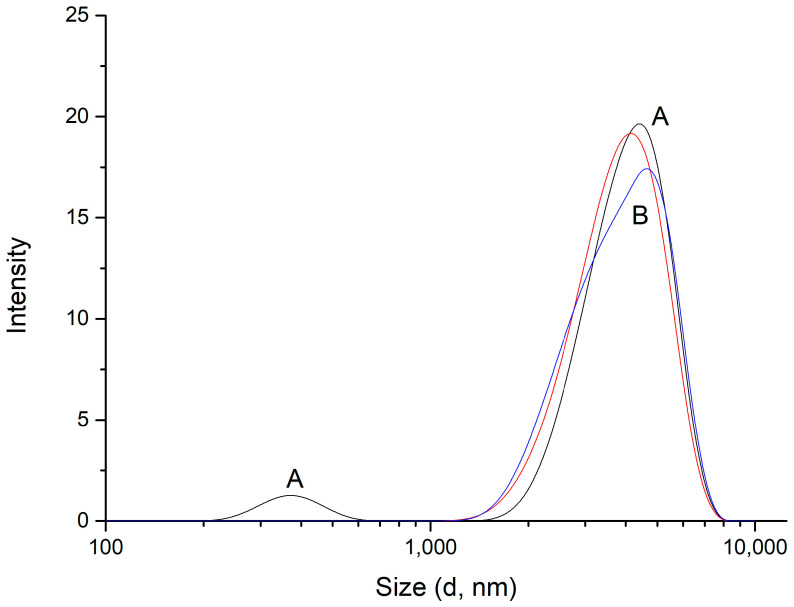
DLS measurements for (A) PEC (black) and (B) PEC formed in presence of **3** (red) and **4** (blue).

**Figure 8 polymers-16-03348-f008:**
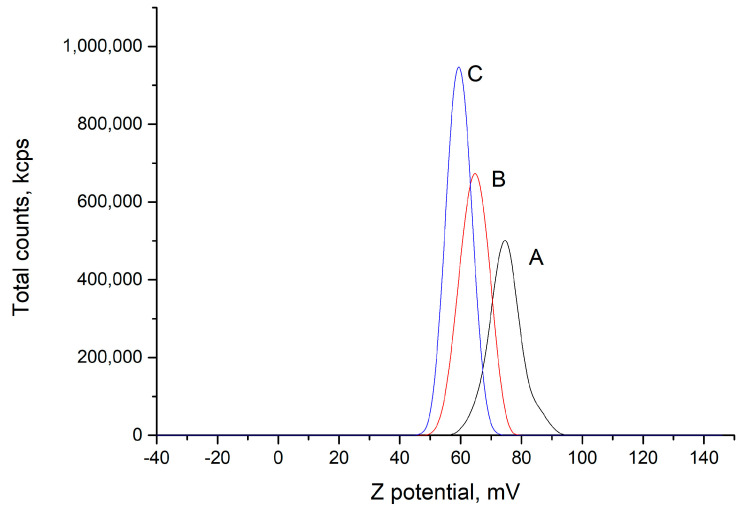
Zeta potential measurements for (A) PEC, (B) PEC formed in presence of **3**, (C) PEC formed in presence of **4**.

**Figure 9 polymers-16-03348-f009:**
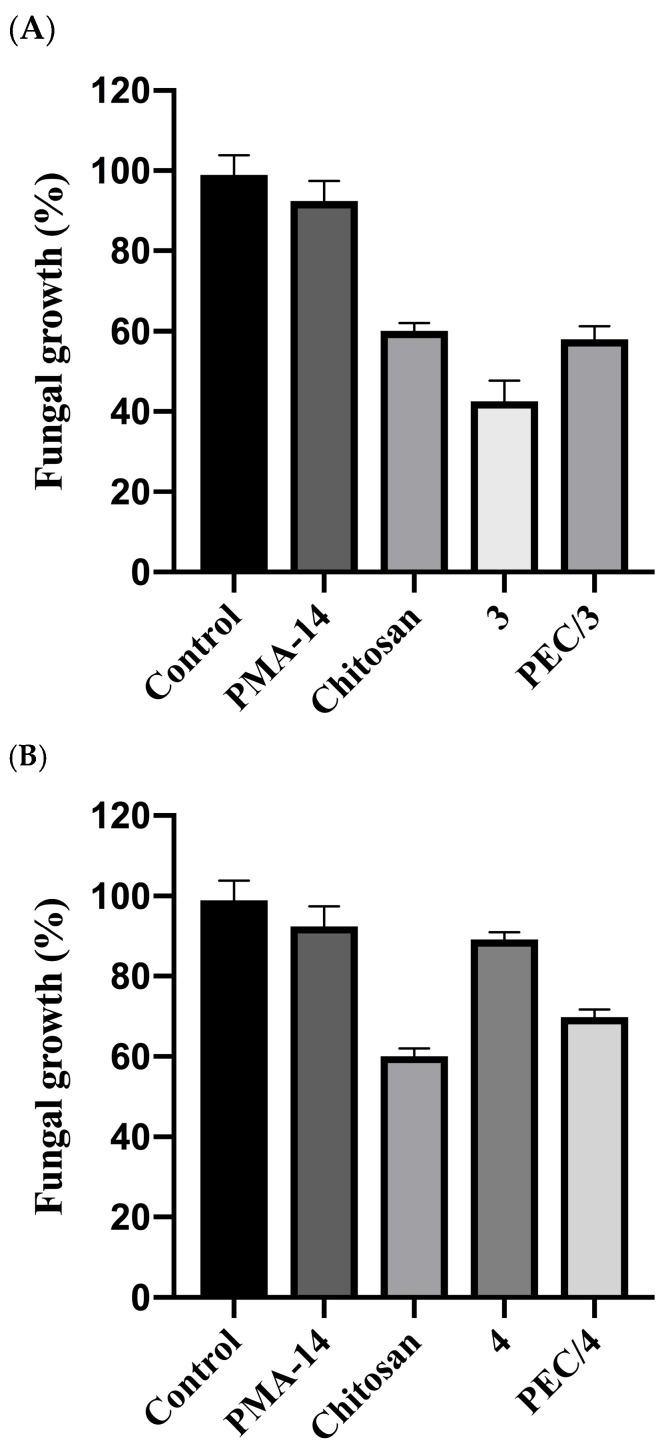
Mycelial growth of *B. cinerea* in presence of (**A**) derivative **3** in free and encapsulated forms (PEC/3) and polyelectrolytes; (**B**) derivative **4** in free and encapsulated forms (PEC/4) and polyelectrolytes.

**Figure 10 polymers-16-03348-f010:**
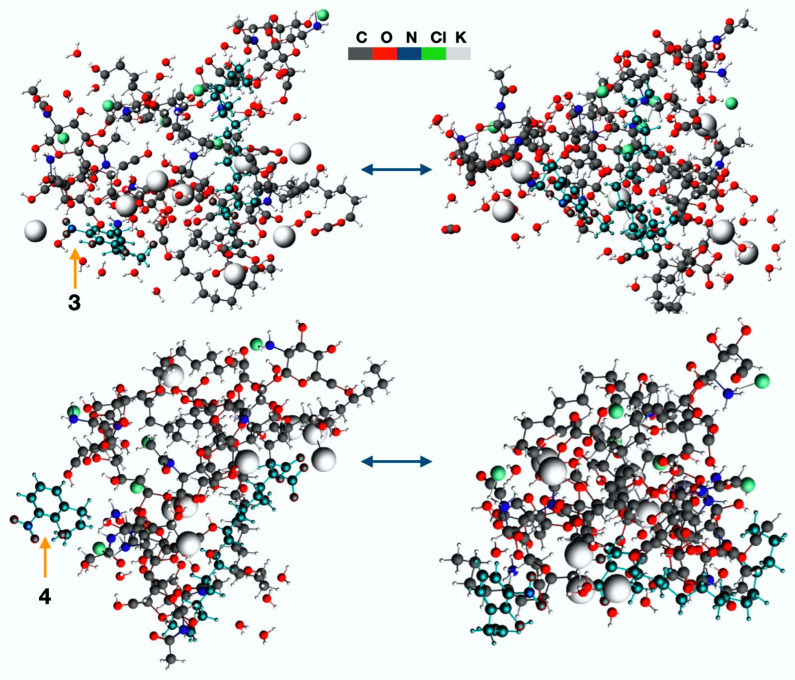
Molecular structures of the initial and final steps obtained in the molecular dynamic simulations. Interactions between polymers and derivatives **3** and **4** are shown.

**Table 1 polymers-16-03348-t001:** Energies obtained from MD simulation, showing the gap between the final and initial step of the simulations (E_I_: Initial Energy and E_F_: Final Energy).

	E_I_ (kcal·mol^−1^)	E_F_ (kcal·mol^−1^)	ΔE_F-I_ (kcal·mol^−1^)
PEC/**3**	−63,532	−63,766	−234
PEC/**4**	−71,811	−71,942	−131

## Data Availability

Data are contained within the article.

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
