# Peer review of "Formation and Characterization of Chitosan-Based Polyelectrolyte Complex Containing Antifungal Phenylpropanoids"

_polymers, 2024, doi:10.3390/polym16233348_

Round 1

Reviewer 1 Report

Comments and Suggestions for Authors

The manuscript is dedicated to the current direction of research in obtaining and applying polyelectrolyte complexes as carriers for the encapsulation of biologically active substances, which can be utilized in various fields of human activity. However, the text is difficult to comprehend and understand due to the use of general "beautiful" phrases rather than specific information. It appears that the authors utilized computer technologies (artificial intelligence) to create the article, but they did not edit the verbose text.

Here are some key observations:

1. After each general statement in the introduction, there are references (3-4 cited sources) (lines 44, 45, 47, 48, 50-51, and 52). These references are mostly unrelated to the research topic.

2. Furthermore, reference 26 does not correspond to what the authors are writing about in the manuscript (line 63).

3. The information provided in the Introduction does not assist in understanding the purpose of the study.

1. The study discusses the reasons why the authors selected these specific polymers.

2. It should be noted that there are some issues with the design of the paper. For example, in Section 2, the subsections are not clearly numbered.

3. I suggest that the authors could improve the clarity of their reasoning by providing a schematic illustration of their ideas on pages 6-7 (lines 211-244).

4. The authors used chitosan and PMA-14 in a 2:1 ratio to create PEC, but the zeta potential did not change significantly compared to chitosan particles. Could the authors explain how they expect this to affect the effectiveness of PEC for drug delivery?

5. The discussion of the results contains a large number of hypotheses that the authors expect to verify in future studies. Perhaps it would be more appropriate to publish the results after the hypotheses have been verified.

6. Based on what evidence did the authors conclude that PECs hold promise for drug delivery (lines 407-409)?

Author Response

Comment 1:

  1. After each general statement in the introduction, there are references (3-4 cited sources) (lines 44, 45, 47, 48, 50-51, and 52). These references are mostly unrelated to the research topic.

 Response 1:

We have checked all mentioned references and we eliminated reference 16. All others are related to the previous sentence.

Comment 2:

  1. Furthermore, reference 26 does not correspond to what the authors are writing about in the manuscript (line 63).

Response 2:

In this case, the reviewer was right and therefore, we have changed that reference.

Cooment 3:

  1. The information provided in the Introduction does not assist in understanding the purpose of the study.

Response 3: 

We have rewritten the Introduction and hope that this version would be more clear for readers. All changes are highlighted.

Comment 4:

  1. The study discusses the reasons why the authors selected these specific polymers.

Response 4:

We agree with this point and therefore we have reorganized the manuscript adding a paragraph explaining the advantages of using chitosan and PMA-14. Also, new references have been added. Lines 90 – 97.

Comment 5:

  1. It should be noted that there are some issues with the design of the paper. For example, in Section 2, the subsections are not clearly numbered.

Response 5:

Done

Comment 6:

  1. I suggest that the authors could improve the clarity of their reasoning by providing a schematic illustration of their ideas on pages 6-7 (lines 211-244).

Response 6:

We have included a new figure that gives a clearer idea on how pyrene fluorescence provides information regarding its environment.

Comment 7:

  1. The authors used chitosan and PMA-14 in a 2:1 ratio to create PEC, but the zeta potential did not change significantly compared to chitosan particles. Could the authors explain how they expect this to affect the effectiveness of PEC for drug delivery?

Response 7:

The similar values obtained for z potential of chitosan and PEC are discussed in lines 279-284.

Comment 8:

  1. The discussion of the results contains a large number of hypotheses that the authors expect to verify in future studies. Perhaps it would be more appropriate to publish the results after the hypotheses have been verified.

Response 8:

We appreciate the comment. However, as stated in lines 446-448, we noted that "These findings highlight the importance of a deeper understanding of the relationship between the chemical structure of guest compounds and encapsulation efficiency and effective delivery." We believe that this initial approach could be highly valuable for understanding the structural characteristics of PEC.

Comment 9:

  1. Based on what evidence did the authors conclude that PECs hold promise for drug delivery (lines 407-409)?

Response 9:

This sentence was modified. Now it reads “These results suggest that a deeper comprehension of parameters determining the relationship between encapsulation efficiency and delivery kinetics is needed”

Reviewer 2 Report

Comments and Suggestions for Authors

In this paper, the authors have reported on the preparation of novel polyelectrolyte complex (PEC) composed of chitosan and potassium salt of poly(maleic anhydride-alt-tetradecene). Two nitroeugenol derivatives: 4-allyl-2-methoxy-6-nitrophenol and 2-allyl-6-nitrophenol were successfully encapsulated in the formed PEC. The obtained PEC systems containing the nitroeugenol derivatives were characterized by fluorescence probing techniques, dynamic light scattering, and zeta potential. The antifungal assay of the two nitroeugenol derivatives and nitroeugenol derivatives-containing PECs was also performed. Molecular dynamic simulations were carried out to clarify the interactions between the constituent polyelectrolytes and the active compounds. I recommend this paper for publication in Polymers after minor revision.

I would like to recommend some revisions before further consideration.

1) In the introduction, please briefly discuss the advantages of chitosan and why it was selected for incorporation into polyelectrolyte complex. Additionally, add some citations from the recent years in the examples of polyelectrolyte complexes.

2) Please give not only the name of producer but also town and country for all used chemicals (2. Materials and Methods, p. 3).

3) In the section 3.3. Diffraction light scattering (DLS) and zeta potential (ζ) (3. Results and Discussion) you discuss only the DLS and zeta potential measurements for PEC formed in presence of compound 3. You should to present the changes in DLS and zeta potential for formation of PEC in the presence of compound 4, and also to compare the obtained results for formation of PEC in the presence of both studied compounds.

4) Please revise the last sentence of the conclusions (p. 12), because your chitosan-based PEC system exhibit antifungal activity, but the same PEC system containing phenylpropanoids does not display the enhanced antifungal activity. 

Author Response

Comment 1:

1) In the introduction, please briefly discuss the advantages of chitosan and why it was selected for incorporation into polyelectrolyte complex. Additionally, add some citations from the recent years in the examples of polyelectrolyte complexes.

Response 1:

Done. Lines 90-125.

Comment 2:

2) Please give not only the name of producer but also town and country for all used chemicals (2. Materials and Methods, p. 3).

Response 2:

Done

Comment 3.

3) In the section 3.3. Diffraction light scattering (DLS) and zeta potential (ζ) (3. Results and Discussion) you discuss only the DLS and zeta potential measurements for PEC formed in presence of compound 3. You should to present the changes in DLS and zeta potential for formation of PEC in the presence of compound 4, and also to compare the obtained results for formation of PEC in the presence of both studied compounds.

Response 3:

The figures (DLS zeta and potential (ζ)) were incorporated into the manuscript.

Lines 294-296 and 311-314 were incorporated and highlighted.

Comment 4:

4) Please revise the last sentence of the conclusions (p. 12), because your chitosan-based PEC system exhibit antifungal activity, but the same PEC system containing phenylpropanoids does not display the enhanced antifungal activity. 

Response 4:

This statement is correct. In our study, we observed that compounds 3 and 4 prefer to be within the PEC rather than outside of this medium. For this reason, we emphasize the need for a deeper investigation into the relationship between PEC and the active compound to be encapsulated. This study may also help to better understand the intricate relationship between the components of a PEC and the PEC's interaction with the target compound for encapsulation.

Round 2

Reviewer 1 Report

Comments and Suggestions for Authors

Comments in the file.

Author Response

Comments 1(2):

Regarding comment 1, I understand that you chose not to revise the introduction. This is your right as authors, but I am still not satisfied with your response. I believe it is important to provide a more thorough analysis of the existing literature and justify the need for your study.

Comments 3-4(2):

With regard to comments 3 and 4, the introduction remains largely unchanged. While you have added information about chitosan, it is not clear how this information relates to the specific aims of your study. Additionally, the number of references has increased, but I believe that it is essential for readers to be able to understand the context and significance of your research without having to search for additional information from the references. I hope you will consider these comments carefully and revise your introduction accordingly.

 Response for comments 1 and 3-4:

Thank you for your comments. In response, we have revised the introduction to clarify how the added information about chitosan relates directly to the specific aims of our study. We have worked to ensure that the context and significance of our research are now more clearly articulated so that readers can better understand the study. Paragraphs incorporated are highlighted.

Comments 7(2):

The authors have not addressed the question of why the 2:1 ratio of cationic to anionic polymers results in virtually no change in the charge of the complex. Their explanation about the interaction of PMA-14 with only equatorial amino groups seems to be incomplete and does not fully address this issue.

Response:

Thank you for pointing out this issue. We have incorporated a new paragraph which is highlighted. This additional text provides a detailed explanation of the charge behavior in the PECs. We believe this revision clarifies the matter and enhances the manuscript.

Comments 8(2):

The authors failed to convince me of their point of view on this issue. I look forward to seeing further improvements in the next version of your manuscript

Response:

Thank you again for your comments. We deeply appreciate all your comments, as they help us refine and strengthen the manuscript.
